# The Potential of Mesenchymal Stem/Stromal Cell Therapy in Mustard Keratopathy: Discovering New Roads to Combat Cellular Senescence

**DOI:** 10.3390/cells12232744

**Published:** 2023-11-30

**Authors:** Mohammad Soleimani, Arash Mirzaei, Kasra Cheraqpour, Seyed Mahbod Baharnoori, Zohreh Arabpour, Mohammad Javad Ashraf, Mahmood Ghassemi, Ali R. Djalilian

**Affiliations:** 1Eye Research Center, Farabi Eye Hospital, Tehran University of Medical Sciences, Tehran 1336616351, Iran; msolei2@uic.edu (M.S.); drarashmirzaei@gmail.com (A.M.); cheraqpourk@gmail.com (K.C.); 2Department of Ophthalmology and Visual Sciences, University of Illinois, Chicago, IL 60612, USA; sbahar2@uic.edu (S.M.B.); arabpourzohreh@gmail.com (Z.A.); mashra5@uic.edu (M.J.A.); ghassemi@uic.edu (M.G.); 3Cornea Service, Stem Cell Therapy and Corneal Tissue Engineering Laboratory, Illinois Eye and Ear Infirmary, 1855 W. Taylor Street, M/C 648, Chicago, IL 60612, USA

**Keywords:** mesenchymal stem/stromal cells, MSCs, intrastromal, mustard, nitrogen mustard, mustard keratopathy, senescence, treatment, ocular surface

## Abstract

Mesenchymal stem/stromal cells (MSCs) are considered a valuable option to treat ocular surface disorders such as mustard keratopathy (MK). MK often leads to vision impairment due to corneal opacification and neovascularization and cellular senescence seems to have a role in its pathophysiology. Herein, we utilized intrastromal MSC injections to treat MK. Thirty-two mice were divided into four groups based on the exposure to 20 mM or 40 mM concentrations of mustard and receiving the treatment or not. Mice were clinically and histopathologically examined. Histopathological evaluations were completed after the euthanasia of mice after four months and included hematoxylin and eosin (H&E), CK12, and beta-galactosidase (β-gal) staining. The treatment group demonstrated reduced opacity compared to the control group. While corneal neovascularization did not display significant variations between the groups, the control group did register higher numerical values. Histopathologically, reduced CK12 staining was detected in the control group. Additionally, β-gal staining areas were notably lower in the treatment group. Although the treated groups showed lower severity of fibrosis compared to the control groups, statistical difference was not significant. In conclusion, it seems that delivery of MSCs in MK has exhibited promising therapeutic results, notably in reducing corneal opacity. Furthermore, the significant reduction in the β-galactosidase staining area may point towards the promising anti-senescence potential of MSCs.

## 1. Introduction

Mustard gas (MG) serves as a potent blistering and alkylating agent with a substantial history of utilization in warfare and terrorist activities. The eyes are particularly vulnerable and are often a preferred target of MG, exhibiting a wide range of possible damages. Mustard keratopathy (MK) is identified as the most prevalent form of eye affliction due to MG, presenting various ocular surface abnormalities, including limbal ischemia, irregularity of the cornea, and detachment between epithelial and stromal layers, along with corneal thinning, neovascularization, and opacity. Mustard agent has two chemical analogues: nitrogen mustard (NM) and sulfur mustard (SM) [1].

NM has been identified to have mutagenic and cytotoxic effects and can indirectly result in single- or double-strand breaks in DNA [2,3,4]. Experiments on rabbits exposed to NM have revealed symptoms such as apoptosis, separation of the epithelial–stromal layer, degradation of corneal collagen, and neovascularization [1,3,5,6,7]. Furthermore, extensive molecular and cellular studies employing murine corneal injury models and rabbit corneal culture models have demonstrated that exposure to NM escalates apoptotic cell death, cyclooxygenase 2, inflammatory mediations, matrix metalloproteinase-9, inflammatory markers, vascular endothelial growth factor, and angiogenic factors, and induces oxidative stress [1,8,9,10,11]. Recent studies have enriched the understanding of mustard keratopathy (MK) physiopathology by identifying cellular senescence as an additional factor contributing to this condition [4]. Managing MK remains highly challenging, with no universally accepted or standardized protocol available currently for its treatment.

Mesenchymal stem/stromal cells (MSCs), notable for their self-renewing and multipotent characteristics, can differentiate into various cell lineages, including epithelial cells, and are easily obtainable from tissues like bone marrow or adipose tissue [12,13,14]. Due to their anti-inflammatory, immunomodulatory, and substantial antioxidant properties, MSCs have become a widely used therapeutic option, protecting cells from apoptosis, vascular and oxidative DNA damage, and cell death [15,16,17]. Particularly in ocular tissues, these cells exhibit anti-inflammatory and anti-fibrotic effects, serving as a viable option for corneal wound healing [18,19,20].

Several methodologies for delivering MSCs to the injured cornea have been introduced, encompassing intrastromal and subconjunctival injections of MSCs, and the grafting of MSCs onto an amniotic membrane which is then sutured to the cornea [21,22,23,24,25,26]. Our recent findings indicate that, while both intrastromal and subconjunctival approaches are safe and effective, the intrastromal method holds a superiority in accelerating healing and reducing post-injury opacification and neovascularization in a murine model of limbal stem cell deficiency (LSCD) [27].

Given the remarkable properties of MSCs, including their ability to differentiate into diverse cell types and their promising outcomes in prior applications for treating ocular surface diseases, we embarked on testing a novel hypothesis regarding the effectiveness of human-bone-marrow-derived MSCs (hBM-MSCs) in accelerating the healing of mustard-induced keratopathy and exploring their potential role in addressing cellular senescence. Consequently, a mouse model was developed for NM-induced keratopathy, where MSCs were injected intrastromally. This study aimed to assess both the clinical and pathocellular implications of MSC therapy on this condition.

## 2. Materials and Methods

### 2.1. Preparation of hBM-MSCs

The Health Sciences Institutional Review Board of the University of Illinois in Chicago granted ethical approval and consent for the collection of bone marrow samples. Human Bone Marrow Mesenchymal Stem/Stromal Cells (hBM-MSCs) were procured using an enzymatic technique. In summary, bone marrow tissue underwent application of 2.5 mg/mL of collagenase I, followed by incubation in MEM-Alpha medium at 37 °C for a duration of 45 min. Subsequently, it was introduced to phosphate-buffered saline (PBS) and underwent centrifugation at 2000 rpm for a period close to 5 min. Upon isolation, the cells were located in a culture flask, coupled with an appropriate culture medium. This medium comprised MEM-Alpha medium, 10% FBS, and 1% penicillin/streptomycin and was renewed two times weekly. Post the third passage, preparations were made for the cells to be used for identification and further investigative studies in cell culture. The cells exhibited characteristics that were aligned with the International Society for Cellular Therapy’s guidelines. We nurtured the cells in a medium that contained serum (MEM-Alpha + 10% fetal bovine serum, 1X L-glutamine, 1X NEAA), created by Corning, Manassas, VA, USA. During the processing, multiple washings were carried out to eliminate xenogeneic components. Post-aliquoting of the MSCs, the turbid vials were subject to cryopreservation in CryoStor^®^ CS5 Cell Freezing Medium’s freezing container. These vials were then situated in a liquid nitrogen tank to maintain them in the vapor phase and were conserved overnight at −80 °C. To evade any detrimental impacts on the cellular morphology and function, the cells were cryopreserved for a maximum duration of six months before being deployed for in vivo experiments.

### 2.2. Immunophenotyping of MSCs

Flow cytometry was performed to evaluate the expression of surface markers in hBM-MSCs, with the analytical process including a sample size consisting of three units. Cells were suspended in a buffer containing PBS and 2 mM of EDTA, with a concentration set at 10^6^ cells/mL. Following this, aliquots of 50 μL (5 × 10^4^ cells/μL) of the cells were relocated into tubes designed for flow cytometry. These were then incubated at a temperature of 4 °C for a duration of 15 min, along with specific monoclonal antibodies which included anti-human CD73, CD90, CD34, andCD45. The procedure for negative control staining involved the usage of FITC-conjugated human antibodies, with all of them sourced from BD Biosciences. Post this phase, the cells were subjected to washing with PBS and then diluted in 500 μL of PBS integrated with 2 mM EDTA. The array of fluorescent dyes that were employed during this analytical assessment incorporated fluorescein isothiocyanate (FITC).

### 2.3. NM Preparation

The liquid NM (mechlorethamine hydrochloride), acquired from Sigma-Aldrich, St. Louis, MO, USA, was carefully preserved at a temperature of −80 degrees Celsius until the designated day of the study. To prepare varying concentrations, NM was diluted using PBS. For the purposes of this study, concentrations of 20 mM and 40 mM were specifically utilized.

### 2.4. Animals

Mice belonging to the C57BL/6J strain, aged between six and eight weeks, were incorporated into this study. Every mouse was managed in compliance with the ARRIVE (Animal Research: Reporting of In Vivo Experiments) guidelines to ensure ethical and accurate handling and reporting. The Biosafety office of the University of Illinois, Chicago gave their approval and maintained supervision over all the protocols and experimental procedures undertaken during the study. The mice were housed under controlled diurnal cycles, consisting of 12 h of light and 12 h of darkness, and were provided with unrestricted access to food.

### 2.5. Study Arms

Mice were divided into four groups: A, B, C and D. Groups A and B received a concentration of 20 mM of NM and groups C and D received a 40 mM concentration of NM. On the first day after NM exposure, groups B and D were treated with an intrastromal injection of MSCs (100,000 cells/5 µL) (treatment arm), while groups A and C were treated with intrastromal injection of the same amount of freezing medium (control arm). All mice, regardless of the study arm, received daily Neomycin and Polymyxin B sulfates for a week.

### 2.6. NM Exposure and MSC Injection

Prior to exposure to NM, the mice were anesthetized using a combination of 100 mg/kg ketamine and 10 mg/kg xylazine. Subsequently, the corneas of the mice were exposed to 5 μL of the predetermined concentration of NM for a duration of 5 min, which was followed by extensive washing with PBS. Mice demonstrating severe signs such as pronounced eye swelling, the formation of pus or ulcers, a reduction in food and water consumption, weight loss, accelerated breathing, and the occurrence of teeth grinding were systematically excluded from the study.

On the day following NM exposure, mice in both the treatment and control groups received intrastromal injections. Specifically, 100,000 cells/5 µL of hBM-MSCs were administered to the treatment group, and Cryostor5 freezing medium was administered to the control group. These injections were performed in the periphery of the cornea using a 32-gauge needle. This technique has been recently and safely trialed by our research team [25]. Additionally, any mice that experienced corneal perforation during the injection process were also excluded from the study.

### 2.7. Clinical Assessments

All included mice underwent evaluations on the day following NM exposure, immediately prior to MSC injection, and subsequently on day 7. Then, evaluations were conducted weekly for a duration of one month and thereafter, monthly, extending up to four months. Any manifestation of corneal opacity and neovascularization were meticulously recorded and imaged by a specialized cornea surgeon at every assessment. These images were then transferred to ImageJ software (version 1.53q) (http://imagej.nih.gov/ij/, accessed on 30 March 2022; provided in the public domain by the National Institutes of Health, Bethesda, MD, USA), enabling the quantification of areas demonstrating corneal opacity and neovascularization. The corneal area underwent manual segmentation with the mean pixel value being automatically deduced. All procedural measurements, experiments, interpretations, and subsequent analyses were conducted in a blinded manner.

### 2.8. Histopathological Assessments

Post the completion of four months, euthanasia was performed on all the mice through the inhalation of carbon dioxide (CO_2_). Corneal samples were then procured and prepared for detailed histopathological evaluations, which included staining processes utilizing hematoxylin and eosin (H&E, Baton Rouge, LA, USA) to determine the degree of fibrosis and beta-galactosidase (β-gal) staining for the identification of senescent cells [18]. An expert pathologist, blind to the sample labeling, assigned fibrosis severity scores as follows: 0 for none; 1 for mild; 2 for moderate; and 3 for severe. Additionally, the phenotype of the regenerated corneal epithelium was assessed and documented through immunocytochemistry utilizing CK12.

### 2.9. Statistical Analysis

All statistical analyses were completed using SPSS (version 22.0, IBM Corp., Armonk, NY, USA) and Excel (2013, Microsoft Inc., Redmond, WA, USA). Results are presented as mean ± standard deviation (SD). The normality of the data was tested using the D’Agostino and Pearson normality test. Based on the normality test, the Mann–Whitney U-test or 2-sided student’s *t*-test and also Fisher’s exact test were performed to determine significance; *p*-value < 0.05 was considered statistically significant.

## 3. Results

### 3.1. Immunophenotyping of MSCs

Flow cytometry analysis was conducted to assess the surface marker expression in hBM-MSCs with a sample size of *n* = 3. The results revealed that hBM-MSCs exhibited a remarkable positivity of over 95% for CD90 and 91% for CD73 MSC surface markers. Conversely, they tested negative for CD34, CD45, and human hematopoietic stem cell markers (Figure 1).

### 3.2. Clinical Results

In total, 32 mice were included in the study, with 8 mice in each group. Figure 2 provides clinical samples of each group. Both groups exposed to a 20 mM concentration of NM (groups A and B) had lower corneal opacity and vascularization than the 40 mM-exposed groups (groups C and D) in the fourth-month visit. In the final follow-up visit, although vascularization was lower in the groups that received MSCs compared to control groups with the same exposure to NM, the difference was not statistically significant (3.88 ± 0.98 to 3.96 ± 1.01 for 20 mM concentration and 6.15 ± 1.27 to 6.65 ± 1.13 for 40 mM concentration with *p*: 0.89 and 0.55, respectively) (Figure 3). However, a statistically significant difference was detected in corneal opacity four months after exposure to 20 and 40 mM NM (2.74 ± 0.98 to 4.23 ± 1.45 for 20 mM concentration and 13.30 ± 1.48 to 23.32 ± 2.24 for 40 mM concentration with *p*: 0.01 and 0.0002, respectively) (Figure 4).

### 3.3. Histopathological Results

Figure 5 provides clinical samples of each group. Similar to the clinical results, both groups exposed to a 20 mM concentration of NM (groups A and B) had lower β-gal staining than the 40 mM exposed groups (groups C and D) in the fourth-month visit. Notably, the β-gal staining area was significantly lower in the groups that received MSC compared to control groups with the same exposure of NM (2.86 ± 0.8 to 4.79 ± 1.61 for concentration of 20 mM and 9.16 ± 1.33 to 11.81 ± 2.37 for concentration of 40 mM with *p*: 0.03 and 0.02, respectively) (Figure 6). In addition, although the treated groups showed a lower severity of fibrosis compared to the control groups, *p* values were not significant (*p*: 0.28 for 20 mM and *p*: 0.99 for 40 mM) (Table 1). CK12 staining, a marker of corneal epithelial regeneration, was more prominent in groups that received treatment (Figure 5).

## 4. Discussion

The toxicity of mustard is a well-established phenomenon known to inflict damage on various bodily organs such as the eyes, liver, kidneys, and spleen [1,28]. The cornea is notably one of mustard’s preferred targets, a preference potentially attributed to its exposure to external environments and the highly replicative nature of epithelial cells [29]. The array of clinical manifestations induced by mustard in the cornea includes edema, opacity, thinning, ulceration, and neovascularization, all of which are collectively termed as mustard keratopathy (MK). Generally, the presentations of MK are categorized into acute, chronic, and delayed-onset forms. The pathophysiology underlying MK is intricate and stems from multiple factors. It encompasses the alkylation of nucleic acids and proteins, the induction of apoptosis, oxidative stress, and inflammation among other contributing factors. Importantly, through a recently conducted murine model, we have gained novel insights into the pathophysiology of chronic/delayed-onset MK and have added senescence induction to the list of contributing elements [4]. We discerned a significant association between β-gal staining, a validated biomarker of senescence, and the concentration of NM, indicating a dose-dependent process of senescence induced by NM. Studies have provided evidence for the therapeutic effectiveness of dexamethasone, doxycycline, silibinin, and a derivative of human fibroblast growth factor-1 in improving corneal injuries induced by NM. Nevertheless, exposure to NM has the potential to progress into a chronic or delayed-onset mustard keratopathy (MK) characterized by substantial visual impairment and a multitude of corneal pathologies. Despite the relentless pursuit to devise effective remedies, a universally acknowledged and standardized treatment to mitigate or reverse MK remains elusive, making the medical management of affected patients exceedingly complex [29].

Cellular senescence is a condition that causes irreversible growth arrest and is distinguished from normal metabolism by a variety of phenotypic changes. The end of a cell’s replicative or Hayflick limit is traditionally when senescence is initiated. This type of senescence, which is a normal part of aging, was initially thought to be caused by the telomere shortening that happens with every cell division [30]. However, a more comprehensive understanding of senescence as a cellular algorithm with multiple triggers has replaced this linear, unidirectional relationship between cell division and senescence. Numerous stressors, such as oncogene activation, oxidative stress, radiation, mitochondrial injury, and inflammation, can prematurely activate senescence [31,32]. Senescent cells have a wide range of effects because, depending on the situation, they either support or disrupt homeostasis. Senescence has been linked to a number of disease states such as diabetes, atherosclerosis, and cancer. However, research connecting senescence to the pathogenesis of ocular diseases is still in its infancy. Graft-versus-host disease (GVHD), fuchs endothelial dystrophy (FED), dry eye disease, and conjunctivochalasis are among the ocular conditions that senescence has been accused of in their physiopathology. The overall literature on senescence and ocular disease is growing rapidly, bringing attention toward developing anti-senescence treatment [20].

Wound healing is an inherent physiological response aimed at restoring damage to bodily tissues, encompassing the intricate processes of inflammation, the generation of new tissue, and the subsequent remodeling of existing tissue structures. Throughout the course of wound healing, cellular senescence assumes a prominent role. In the context of corneal wound healing, the phenomenon of fibroblast senescence manifests as an anti-fibrogenic phenotype, characterized by diminished reactivity to FGF2 and platelet-derived growth factor-BB, while concurrently exhibiting an elevated expression of MMP1, MMP3, and MMP13 [33]. Conversely, cellular senescence has the capacity to impede the process of wound healing. An exemplification of this interference is observed in inflammation-mediated cellular senescence, wherein the proliferation and migration of fibroblasts, both of which are indispensable for the formation of new tissue, are significantly diminished. Thus, while transient cellular senescence has the potential to facilitate the reparative processes of tissue restoration, the prolonged presence of senescent cells can considerably hinder this intricate process [34].

Mesenchymal stem/stromal cells (MSCs) are multifaceted, self-renewing stem cells capable of differentiating into various cell types. They can be extracted from diverse tissues, including bone marrow, adipose tissue, and the umbilical cord [23]. MSCs are distinguished by their array of advantageous properties such as anti-inflammatory, anti-angiogenic, anti-apoptotic, and senescence inhibitory activities [28]. These beneficial characteristics have rendered MSCs a compelling treatment alternative for managing keratopathies arising from chemical burns, limbal stem cell deficiency (LSCD), and keratitis. Multiple methods exist for administering MSCs in cases of corneal injury, yet the optimum dose and administration route are still under exploration [23]. In a study, the topical, subconjunctival, intravenous (IV), and intraperitoneal administration of MSCs in a rat model were studied and it was reported that intravenous and subconjunctival administration has the highest efficacy in reducing corneal opacity [35]. However, cells administered intravenously must initially traverse the pulmonary system prior to systemic distribution, a phenomenon termed the pulmonary first-pass effect. It has been shown that a substantial portion of MSCs is entrapped within the lung subsequent to IV administration. This problem arises because of the diameter of MSCs, which ranges from 20 to 30 μm, as it has been observed that the quantity of entrapped cells declines with the administration of a vasodilator. Furthermore, in conjunction with size, it is plausible that endothelial cell adhesion molecules contribute to the entrapment of MSCs within the lung, since a reduction in the number of cells ensnared within the lungs when the CD49d receptor is blocked [36]. Also, the systemic circulation of subconjunctivally injected MSCs following the absorption of cells through conjunctival vasculature is a concern. One of the other local routes of cell delivery is intrastromal injection. In this approach, due to the avascularity of the cornea, the treatment can be directed toward the exact site of injury, which provides a higher density of cells adjacent to the damaged site and decreases the risk of systemic absorption of the cells. In a recent study on a murine model of LSCD, we showed that the intrastromal administration of MSCs provides significantly higher efficacy in the healing process and reducing post-injury opacification and neovascularization of the cornea [27]. We showed that corneal distribution of MSCs on the intrastromal route is significantly higher and MSCs would last up to 1 month in the cornea.

In addition to numerous other beneficial outcomes, stem cells and their extracellular vesicles are capable of exerting senolytic activity. For instance, mesenchymal stem cells derived from bone marrow have been shown to decrease senescence and enhance cardiac function in aged mice [37]. Similarly, mesenchymal stem cells derived from human umbilical cords have been observed to protect rat kidneys from acute kidney injury-induced senescence [38]. Furthermore, extracellular vesicles derived from mesenchymal stem cells have been found to inhibit oxidative stress-induced senescence in endothelial cells, promote wound closure in ageing diabetic mice [39], and decrease myocardial senescence, potentially by improving the systemic inflammatory profile, in a pig model of metabolic renovascular disease [40]. The anti-senescence mechanisms of stem cells and extracellular vesicles have primarily been attributed to their contents, which have the ability to regulate the senescence-associated secretory phenotype, repair damaged organelles, and restore the function of senescent cells. However, it is also likely that stem cells and their extracellular vesicles have indirect effects on senescence by restoring the microcirculation of tissues, enhancing the functionality of parenchymal cells, and inhibiting the activation of immune cells. Therefore, it may be premature to consider their use as primary interventions for combating senescence. Thus far, there have been no reported clinical trials utilizing stem cells or their extracellular vesicles to specifically target senescence [41].

The present report, which was in the continuation of our recent works, studied the application of MSCs in the management of MK. Our study showed that the intrastromal administration of MSCs had promising healing effects on corneal opacification. In addition, β-gal staining decreased significantly, which may indicate a promising role for MSCs in combating senescence. However, the decrease in neovascularization and fibrosis was not statistically significant, which can be related to the small sample size of our study. In line with our results, a recent study investigated the therapeutic potential of MSCs in NM-induced corneal wounds [42]. They added NM to different corneal cells, and the ocular surface of pigs and mice. They found that the utilization of MSC-conditioned media (CM) subsequent to NM exposure had a partial restorative effect on mitochondrial function and resulted in a reduction in intracellular ROS generation, thereby facilitating cell survival. The implementation of MSC-CM therapy exhibited an augmentation of the wound healing process. MSC-CM displayed an inhibitory effect on NM-induced apoptotic cell death in both murine and porcine corneal tissue. The utilization of MSC-CM subsequent to a chemical insult yielded considerable enhancements in the preservation of corneal structure and wound healing. In vitro, ex vivo, and in vivo findings collectively indicate that MSC-CM holds potential as a targeted therapeutic approach for the treatment of chemical eye injuries, including MK [42].

It should be mentioned that, in ophthalmological stem-cell-based studies, the injection of freezing medium is routinely considered as a control. The safety of the freezing medium has been previously documented by our research team, since no adverse effects were observed in corneas that were treated with this substance in our previously published studies with a sufficient period of follow-up. In line with this point, the used freezing medium, CryoStor^®^ CS5, is serum-free, animal-component-free, cGMP-manufactured, and formulated with USP-grade components. Given lack of trophic factors, therapeutical effects and benefits can be linked to MSCs. There exist several limitations in our study. First, our sample size was relatively small. Second, our animal models did not involve a comparative analysis of other administration routes. Third, we did not analyze the other markers of senescence. Fourth, the quantitative assessment of corneal neovascularization and fibrosis using CD31 and LYVE1, respectively, can yield more precise and detailed results. Consequently, future research endeavors are imperative to overcome these limitations and enhance our comprehension of the prospective applications of MSCs in treating subjects exposed to NM.

## 5. Conclusions

The intrastromal delivery of MSCs in a murine model of mustard-induced keratopathy has exhibited promising therapeutic results, notably in reducing corneal opacity. Furthermore, the significant reduction in the β-galactosidase staining area may point towards the promising anti-senescence potential of MSCs.

## Figures and Tables

**Figure 1 cells-12-02744-f001:**
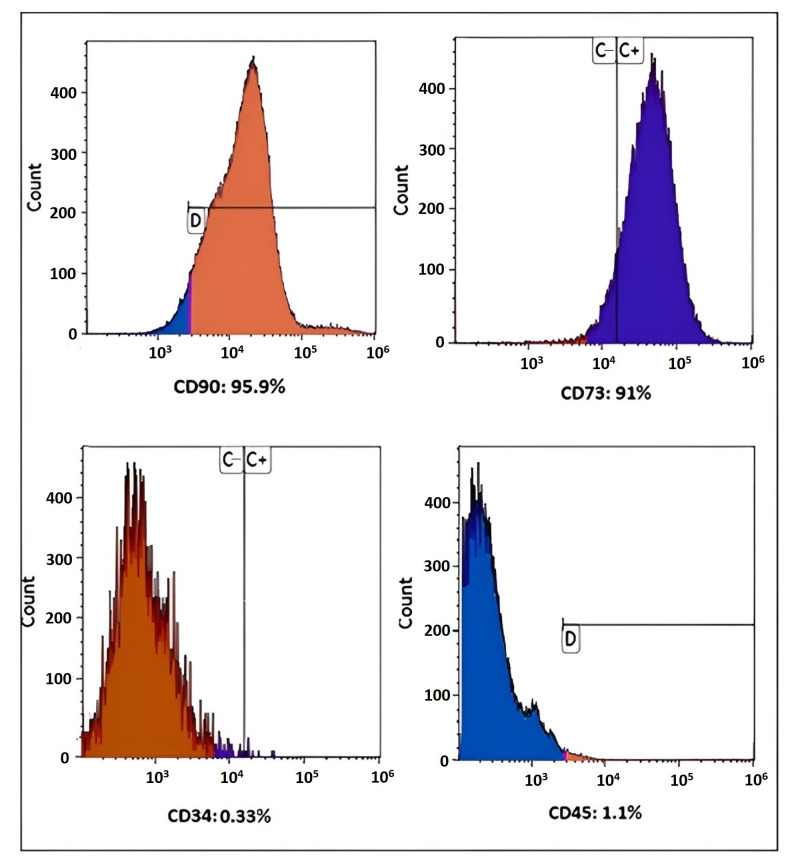
Flow cytometric analysis of hBM-MSC surface markers, showing profiles of CD73, CD90, CD34, and CD45.

**Figure 2 cells-12-02744-f002:**
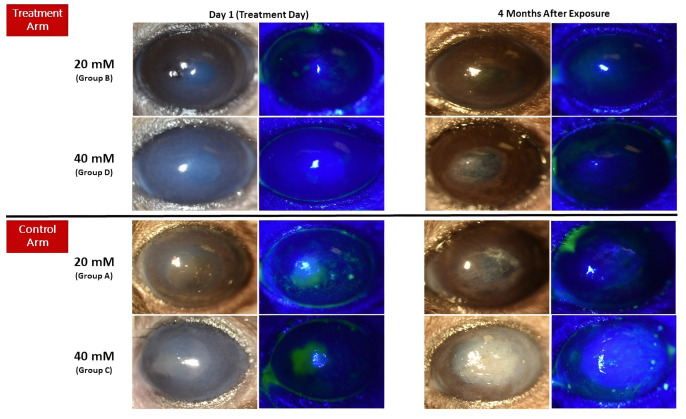
Clinical samples of each group, showing slit-lamp photography and corresponding fluorescein staining. Treated corneas with MSC showed lower opacity compared to non-treated similarly NM exposed corneas.

**Figure 3 cells-12-02744-f003:**
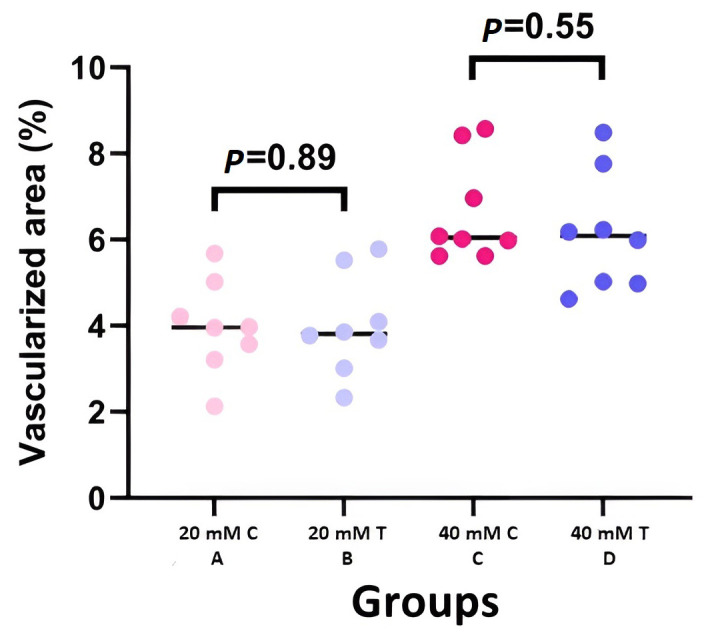
Scatter-dot plot of vascularized area percentage in different groups of study (C: control, T: treatment) (B, D: treatment arm and A, C: control arm). Although vascularization was lower in the groups that received MSCs compared to control groups with the same exposure of NM, the difference was not statistically significant.

**Figure 4 cells-12-02744-f004:**
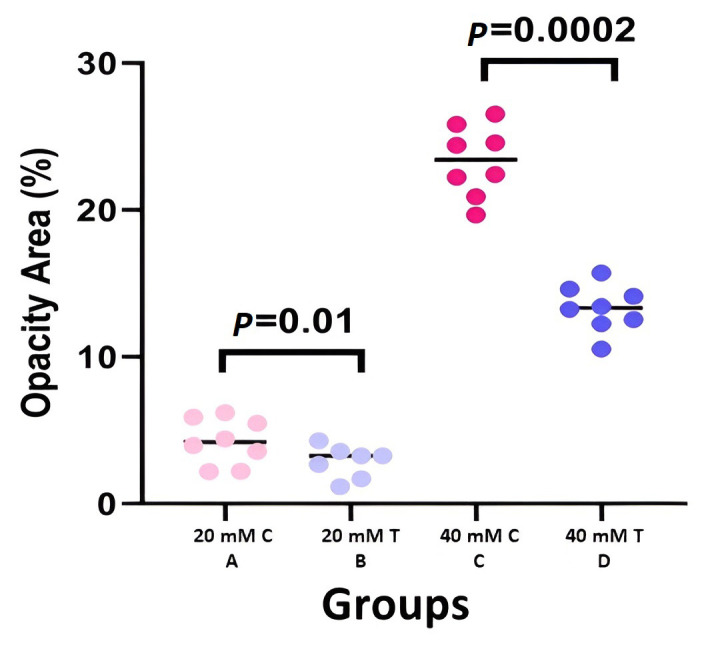
Scatter-dot plot of corneal opacity area percentage in different groups of study (C: control, T: treatment) (B, D: treatment arm and A, C: control arm). A statistically significant difference was detected in corneal opacity four months after exposure to 20 and 40 mM of NM, showing the superiority of MSC treatment.

**Figure 5 cells-12-02744-f005:**
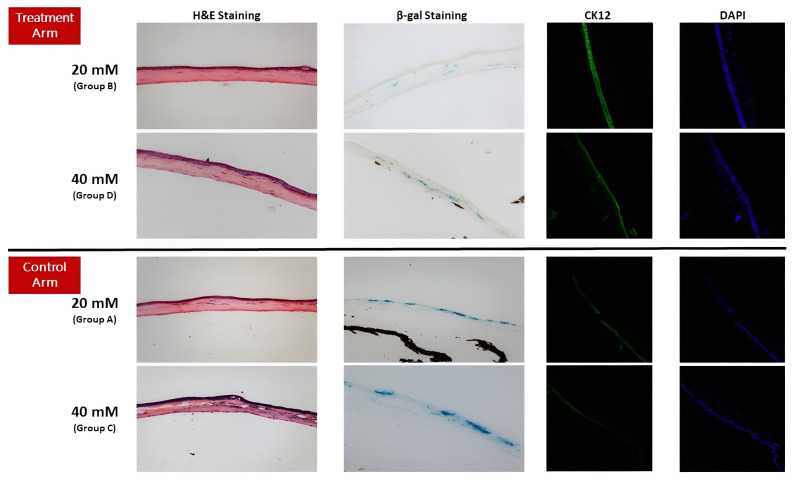
Histopathological samples of each group, showing hematoxylin and eosin (H&E), beta-galactosidase (β-gal), and CK12 staining. Corneas treated with MSC showed lower β-gal staining and fibrosis compared to non-treated, similarly NM exposed corneas. Also, CK12 staining was more prominent in treated groups, proposing a better epithelium regeneration. DAPI shows the sectioned cornea for CK12 staining.

**Figure 6 cells-12-02744-f006:**
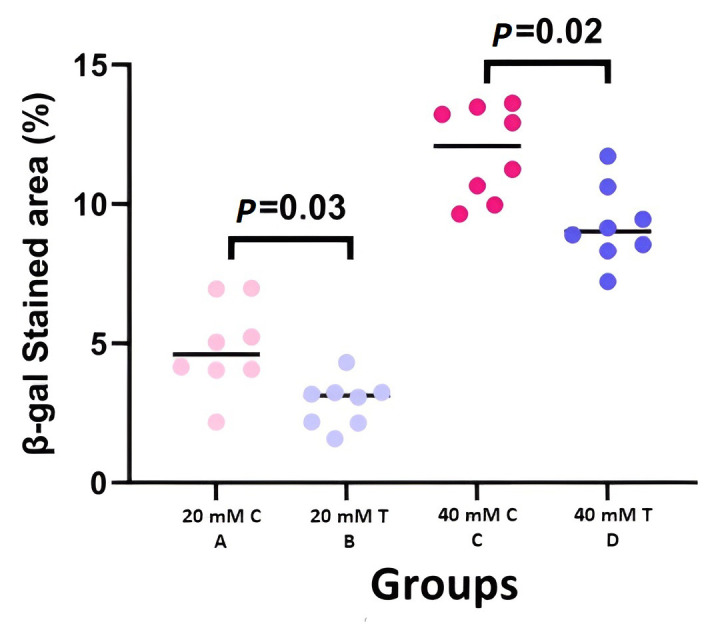
Scatter-dot plot of β-gal staining area percentage in different groups of study (C: control, T: treatment) (B, D: treatment arm and A, C: control arm). The β-gal staining area was significantly lower in the groups that received MSC compared to control groups with the same exposure of NM.

**Table 1 cells-12-02744-t001:** Severity of fibrosis in different groups. Although the treated groups showed lower severity of fibrosis compared to the control groups, *p* values were not significant (*p*: 0.28 for 20 mM and *p*: 0.99 for 40 mM).

	**Control Arm**		**Treatment Arm**	
	20 mM	40 mM	20 mM	40 mM
Grade 0	0	0	0	0
Grade 1	1	0	4	1
Grade 2	6	4	4	6
Grade 3	1	4	0	1

## Data Availability

All the data are available with the corresponding author upon request.

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
