# Peer review of "The Potential of Mesenchymal Stem/Stromal Cell Therapy in Mustard Keratopathy: Discovering New Roads to Combat Cellular Senescence"

_cells, 2023, doi:10.3390/cells12232744_

Round 1
Reviewer 1 Report
Comments and Suggestions for Authors
The authors evaluated the effect of an intrastromal MSC injection on the cornea with nitrogen mustard injury. While the topic is interesting, more experiments would support the authors' conclusion. Here are some suggestions for improving the study.
1. To assess the benefits of MSC therapy for mustard keratopathy more effectively, the authors should consider including a control group that receives no treatment and comparing it with the group treated with MSCs or freezing medium. This would enable readers to determine whether MSC treatment is beneficial compared to no treatment and whether freezing medium itself is toxic to the cornea. In the same vein, to rule out potential adverse effects of the freezing medium, it's advisable to conduct MSC injections in PBS vehicle by reconstituting cells in PBS after thawing. The results would provide a valuable, clinically relevant information on whether MSCs should be administered in PBS or can be administered directly from cryotubes.
2. The study could benefit from including additional outcome measures, especially for the assessment of corneal neovascularization (NV) and fibrosis. The use of markers like CD31 or LYVE1 for assessing corneal NV and collagen staining for fibrosis would yield more precise and detailed results. In the representative photos of the corneas (figure 2), it is hard to notice any NV even in severely damaged corneas. The NV in the corneal stroma can be better assessed by CD31 or LYVE1 staining of corneal sections. In addition, the criteria used to assign fibrosis scores in H&E-stained sections should be provided in detail, and these scores (table 1) should be subject to statistical evaluation to establish statistical significance.
3. Changing the graphs to scatter-dot plots is recommended to enhance data visibility and enable readers to easily interpret individual data points.
4. The reviewer suggests using the term "mesenchymal stem/stromal cells" (MSCs) to accurately describe the cell population being studied because the cells which the authors isolated in this study are mesenchymal stromal cells containing a small population of mesenchymal stem cells. This change adheres to established nomenclature suggested by ISCT.
Author Response
Dear Editor and Reviewers,
We are very grateful for thoughtful comments and recommendations. We wish to express our appreciation for the insightful comments. We have carefully addressed the editor and reviewer’s suggestions and in doing so feel the manuscript is substantially strengthened.
Reviewer #1:
- To assess the benefits of MSC therapy for mustard keratopathy more effectively, the authors should consider including a control group that receives no treatment and comparing it with the group treated with MSCs or freezing medium. This would enable readers to determine whether MSC treatment is beneficial compared to no treatment and whether freezing medium itself is toxic to the cornea. In the same vein, to rule out potential adverse effects of the freezing medium, it's advisable to conduct MSC injections in PBS vehicle by reconstituting cells in PBS after thawing. The results would provide a valuable, clinically relevant information on whether MSCs should be administered in PBS or can be administered directly from cryotubes.
Dear reviewer, thank you so much for your valuable comments; the safety of the freezing medium has been previously documented by our research team, since no adverse effects were observed in corneas that were treated with this substance in our previously published studies with a sufficient period of follow-up. It should be mentioned that in ophthalmological stem cell-based studies, injection of freezing medium is routinely considered as a control. In line with this point, the used freezing medium, CryoStor® CS5, is serum-free, animal component-free, cGMP-manufactured, and formulated with USP-grade components. Given lack of trophic factors, therapeutical effects and benefits can be linked to MSCs. Moreover, to the best of our knowledge, there is lacking evidence regarding your last point about whether MSCs should be administered in PBS or can be administered directly from cryotubes. Hence, this novel idea requires a separate general study on the application of stem cells and was outside the scope of our study.
- The study could benefit from including additional outcome measures, especially for the assessment of corneal neovascularization (NV) and fibrosis. The use of markers like CD31 or LYVE1 for assessing corneal NV and collagen staining for fibrosis would yield more precise and detailed results. In the representative photos of the corneas (figure 2), it is hard to notice any NV even in severely damaged corneas. The NV in the corneal stroma can be better assessed by CD31 or LYVE1 staining of corneal sections. In addition, the criteria used to assign fibrosis scores in H&E-stained sections should be provided in detail, and these scores (table 1) should be subject to statistical evaluation to establish statistical significance.
Thanks for your comment; your point regarding the usage of the aforementioned markers is so appreciated and added to the limitations and recommendations of our study. However, it should be mentioned that the CD31 application requires a whole-mounted cornea, while we have sectionized the studied corneas. So, it was not possible for us to re-evaluate the specimens with CD31. NV was assessed by an expert cornea surgeon using high-quality original images. Also, the degree of fibrosis was qualitatively scored by an expert pathologist in a blind manner. To the best of our knowledge and literature review, no specific criteria are there in this regard. Statistical analysis of Table 1 has been added to the manuscript. Based on fisher's exact test, P values were not significant (P: 0.28 for 20 mM and P: 0.99 for 40 mM).
- Changing the graphs to scatter-dot plots is recommended to enhance data visibility and enable readers to easily interpret individual data points.
Thank you for your comment; the graphs are revised as you wished.
- The reviewer suggests using the term "mesenchymal stem/stromal cells" (MSCs) to accurately describe the cell population being studied because the cells which the authors isolated in this study are mesenchymal stromal cells containing a small population of mesenchymal stem cells. This change adheres to established nomenclature suggested by ISCT.
Thank you very much; the revision has been made.

Reviewer 2 Report
Comments and Suggestions for Authors
Authors aimed to assess the usefulness of BM-MSC in the corneal treatment in mustard keratopathy mice model. I have concerns about the applied methodology. Please find my comments below.
1. MSC are known as multipotent cells. Conducting studies employing them faces the toil of carefull examination of their phenotype. Authors stated that the applied criteria for including cells into the study was at least 75% positive cells for CD73 and CD90 and maximum 3% for CD34 and CD45. Such treshold suggest that the used cells were highly heterogenous. I am not conviced if the applied criteria are sufficient for studies on MSC properties. Moreover, the used MSC should be tested against their plasticity what was not showed by the authors. Why did authors use human cells instead of mouse cells?
2. Line 55 - I recommend supporting this statement with appriopriate original study article reference.
3. Line 102 - authors should verify the mentioned number of cells.
4. Why did authors use the mentioned NM concentrations instead of using more or less concentrated doses?
5. It is not clear how the neovascularization was assessed.
6. The statistics could be described in more detail. For example, what test did authors use for the evaluation of data distribution?
7. The figure 2 is of quite poor resolution. The size of pictures within it makes it illegible. It would be benefitial for the data presentation to reorganize this figure and make it bigger vertically than horizontally.
8. Figure captions are quite laconic. It would be beneficial to provide a comprehensive description of each figure. Potential readers would appreciate the clear description of showed data without the need of seeking the informations in the main text.
9. Line 260 - Certainly not. There are more or less 200 types of cells in human body. What did authors mean mentioning myriad of cells?
10. Line 271 - Cells of 20-30 um of diameter are not tiny.
"(...)an affinity for CD94d". What did the authors mean by this remark? The statement is not celar.
Author Response
Dear Editor and Reviewers,
We are very grateful for thoughtful comments and recommendations. We wish to express our appreciation for the insightful comments. We have carefully addressed the editor and reviewer’s suggestions and in doing so feel the manuscript is substantially strengthened.
Reviewer #2:
- MSC are known as multipotent cells. Conducting studies employing them faces the toil of careful examination of their phenotype. Authors stated that the applied criteria for including cells into the study was at least 75% positive cells for CD73 and CD90 and maximum 3% for CD34 and CD45. Such threshold suggest that the used cells were highly heterogenous. I am not convinced if the applied criteria are sufficient for studies on MSC properties. Moreover, the used MSC should be tested against their plasticity what was not showed by the authors. Why did authors use human cells instead of mouse cells?
Dear reviewer, thanks for your comments; you are completely right and the relevant section had some errors and typos especially on numerical values. The required corrections have been made. Also, after incubation in appropriate differentiation medium (StemPro® Differentiation kits, all from Thermofisher), plasticity of MSCs was evaluated similar to our previous works. The purpose of such animal studies is to translate into human clinical trials. So, we usually work on human-derived MSCs in our researches.
- Line 55 - I recommend supporting this statement with appropriate original study article reference.
Thank you; the proper references have been cited.
*Gu S, Xing C, Han J, et al. Differentiation of rabbit bone marrow mesenchymal stem cells into corneal epithelial cells in vivo and ex vivo. Molecular Vision. 2009;15:99.
*Stanko P, Kaiserova K, Altanerova V, et al. Comparison of human mesenchymal stem cells derived from dental pulp, bone marrow, adipose tissue, and umbilical cord tissue by gene expression. Biomed Pap Med Fac Univ Palacky Olomouc Czech Repub. 2014;158(3):373–377.
*Nieto-Nicolau N, Martín-Antonio B, Müller-Sánchez C, et al. In vitro potential of human mesenchymal stem cells for corneal epithelial regeneration. Regenerative Medicine. 2020;15(3):1409–1426.
- Line 102 - authors should verify the mentioned number of cells.
Thank you very much; done.
Cells were suspended in a buffer containing PBS and 2 mM of EDTA, with a concentration set at 106 cells/mL. Following this, aliquots of 50 μL (5×104 cells/μL) of the cells were relocated into tubes designed for flow cytometry.
- Why did authors use the mentioned NM concentrations instead of using more or less concentrated doses?
Thanks for your comment; it was in line with our previous work. In fact, higher doses were severely toxic, leading to corneal melting and, on the other hand, damage from lower doses was not significant.
- It is not clear how the neovascularization was assessed.
Using the ImageJ software, areas of neovascularization were meticulously segmented on original high-quality images by an expert cornea surgeon. Also, the whole corneal area underwent manual segmentation. Then, the area of neovascularization to the whole cornea was automatically calculated and reported.
- The statistics could be described in more detail. For example, what test did authors use for the evaluation of data distribution?
Normality of the data was tested using D’Agostino & Pearson normality test. Based on normality test, Mann-Whitney U-test or 2-sided student’s t-test was performed to determine significance.
- The figure 2 is of quite poor resolution. The size of pictures within it makes it illegible. It would be beneficial for the data presentation to reorganize this figure and make it bigger vertically than horizontally.
Thank you; a revision has been made.
- Figure captions are quite laconic. It would be beneficial to provide a comprehensive description of each figure. Potential readers would appreciate the clear description of showed data without the need of seeking the information in the main text.
Thank you very much; done. The legends have been robust and detailed.
- Line 260 - Certainly not. There are more or less 200 types of cells in human body. What did authors mean mentioning myriad of cells?
Thanks for your comment; a revision has been made.
Mesenchymal stem cells (MSCs) are multifaceted, self-renewing stem cells capable of differentiating into various cell types.
- Line 271 - Cells of 20-30 um of diameter are not tiny.
"(...)an affinity for CD94d". What did the authors mean by this remark? The statement is not clear.
Thanks; the sentence has been revised.
However, cells administered intravenously must initially traverse the pulmonary system prior to systemic distribution, a phenomenon termed the pulmonary first-pass effect. It has been shown that a substantial portion of MSCs is entrapped within the lung subsequent to IV administration. This problem arises because of the diameter of MSCs, which ranges from 20 to 30 μm, as it has been observed that the quantity of entrapped cells declines with the administration of a vasodilator. Furthermore, in conjunction with size, it is plausible that endothelial cell adhesion molecules contribute to the entrapment of MSCs within the lung, since a reduction in the number of cells ensnared within the lungs when the CD49d receptor is blocked.

Round 2
Reviewer 1 Report
Comments and Suggestions for Authors
The concerns raised by the reviewer have been addressed.
Author Response
Reviewer #1:
- The concerns raised by the reviewer have been addressed.
Dear reviewer, we thank you so much for your valuable comments and support again.

Reviewer 2 Report
Comments and Suggestions for Authors
Authors have improved their manuscript. I am satisfied with the responses. It should be noticed that the photos provided within part of the figures look blurry, it is worthy to change them with high resolution ones.
Author Response
Reviewer #2:
- Authors have improved their manuscript. I am satisfied with the responses. It should be noticed that the photos provided within part of the figures look blurry, it is worthy to change them with high resolution ones.
Dear reviewer, we thank you for your valuable comments and support again. The quality of figures drops during uploading a Word document. We are ready to submit them separately, if the publisher requests. Nevertheless, we tried to raise their resolution.
